# Integrating PointNet-Based Model and Machine Learning Algorithms for Classification of Rupture Status of IAs

**DOI:** 10.3390/bioengineering11070660

**Published:** 2024-06-28

**Authors:** Yilu Shou, Zhenpeng Chen, Pujie Feng, Yanan Wei, Beier Qi, Ruijuan Dong, Hongyu Yu, Haiyun Li

**Affiliations:** 1School of Biomedical Engineering, Beijing Key Laboratory of Fundamental Research on Biomechanics in Clinical Application, Capital Medical University, No. 10, Xitoutiao, Youanmenwai, Fengtai District, Beijing 100069, China; 2Beijing Tongren Hospital, Key Laboratory of Otolaryngology Head and Neck Surgery, Capital Medical University, No. 1, Dongjiaominxiang, Dongcheng District, Beijing 100010, China

**Keywords:** intracranial aneurysms, rupture risk, hemodynamic clouds, PointNet, machine learning, geometrical parameters, hemodynamic parameters

## Abstract

Background: The rupture of intracranial aneurysms (IAs) would result in subarachnoid hemorrhage with high mortality and disability. Predicting the risk of IAs rupture remains a challenge. Methods: This paper proposed an effective method for classifying IAs rupture status by integrating a PointNet-based model and machine learning algorithms. First, medical image segmentation and reconstruction algorithms were applied to 3D Digital Subtraction Angiography (DSA) imaging data to construct three-dimensional IAs geometric models. Geometrical parameters of IAs were then acquired using Geomagic, followed by the computation of hemodynamic clouds and hemodynamic parameters using Computational Fluid Dynamics (CFD). A PointNet-based model was developed to extract different dimensional hemodynamic cloud features. Finally, five types of machine learning algorithms were applied on geometrical parameters, hemodynamic parameters, and hemodynamic cloud features to classify and recognize IAs rupture status. The classification performance of different dimensional hemodynamic cloud features was also compared. Results: The 16-, 32-, 64-, and 1024-dimensional hemodynamic cloud features were extracted with the PointNet-based model, respectively, and the four types of cloud features in combination with the geometrical parameters and hemodynamic parameters were respectively applied to classify the rupture status of IAs. The best classification outcomes were achieved in the case of 16-dimensional hemodynamic cloud features, the accuracy of XGBoost, CatBoost, SVM, LightGBM, and LR algorithms was 0.887, 0.857, 0.854, 0.857, and 0.908, respectively, and the AUCs were 0.917, 0.934, 0.946, 0.920, and 0.944. In contrast, when only utilizing geometrical parameters and hemodynamic parameters, the accuracies were 0.836, 0.816, 0.826, 0.832, and 0.885, respectively, with AUC values of 0.908, 0.922, 0.930, 0.884, and 0.921. Conclusion: In this paper, classification models for IAs rupture status were constructed by integrating a PointNet-based model and machine learning algorithms. Experiments demonstrated that hemodynamic cloud features had a certain contribution weight to the classification of IAs rupture status. When 16-dimensional hemodynamic cloud features were added to the morphological and hemodynamic features, the models achieved the highest classification accuracies and AUCs. Our models and algorithms would provide valuable insights for the clinical diagnosis and treatment of IAs.

## 1. Introduction

Intracranial aneurysms (IAs) are local bulges formed in the walls of cerebral arteries due to abnormal pathology changes. The prevalence in the population is around 3% [1], with an annual rupture rate of 0.95–2% [2,3,4,5]. The incidence of detection is very high; the rate of rupture is very low. Once ruptured, there is a high likelihood of causing subarachnoid hemorrhage, leading to high mortality and disability. Therefore, early diagnosis and intervention can help improve clinical outcomes. The urgent challenge in clinical practice is to develop a reliable approach to assess and predict the risk of IAs rupture.

Many studies investigated the risk factors that lead to IAs rupture [6,7,8,9,10,11,12,13], mainly focusing on morphological and hemodynamic features. Kleinloog et al. [10] recognized irregular aneurysm morphology, aspect ratio (AR), size ratio (SR), bottleneck factor, and height-to-width ratio as risk factors. Nikolic et al. [7] discovered that IAs were more susceptible to rupture when they exhibit an irregular shape, an AR over 1.6, an SR over 1.5, and an angle greater than 135 degrees. Cornelissen et al. [14] simulated the hemodynamics of intracranial aneurysms prior to and after rupture using the Computational Fluid Dynamics (CFD) method. They observed alterations in flow complexity, stability, inflow concentration, and the region of flow impingement. Lv et al. [15] applied univariate and multivariate logistic regression to explore the links between the morphological-hemodynamic pattern and the rupture risk of IAs. They found that IAs were more likely to rupture with noticeably bigger sizes and lower normalized wall shear stress. Fujimura et al. [6] conducted a comparative analysis of the morphological and hemodynamic alterations in IAs prior to and after rupture. They observed that ruptured IAs exhibited a transformation into slender and irregular geometrical shapes, concurrent with an increase in aneurysm volume. The observed morphological modifications were accompanied by statistically significant hemodynamic changes, resulting in low wall shear stress due to stagnant flow. Existing studies have shown that the main geometrical parameters affecting IAs rupture include aneurysm size, shape, growth location, AR, SR, and blood flow inflow angle; hemodynamic parameters primarily consist of wall shear stress (WSS), oscillatory shear index (OSI), wall shear stress gradient, region of flow impingement, and velocity.

In recent years, artificial intelligence has been utilized to help identify intracranial aneurysms and predict their rupture risk [16,17,18,19,20,21,22,23,24,25,26]. Liu et al. [26] proposed a machine learning model on radiological morphological parameters to forecast the stability of aneurysms, and they found that flatness was an important morphological factor for predicting the stability of aneurysms. Heo et al. [19] used data from the Korean National Health Examination to train several machine learning models for predicting the risk of IAs rupture; among them, the scalable tree boosting system (XGB) model obtained the best predictive performance with an AUC of 0.765. Shi et al. [27] created machine learning models to predict the likelihood of minor aneurysms rupturing, employing clinical, morphological, and hemodynamic data. The results indicated that the SVM achieved the highest performance in training and internal datasets, with AUC of 0.88 and 0.91, respectively. Their experiments showed that hemodynamic parameters with large weight in predicting intracranial aneurysm rupture risk mainly included steady flow pattern, concentrated inflow jets, a small flow impingement zone (less than 50%), and the coefficient of change of the oscillatory shear index. Tanioka et al. [28] developed a Random Forest-based classification model of the cerebral aneurysm rupture status, applying morphological variables and hemodynamic parameters, and achieved an accuracy of 78.3%. They found that the projection ratio, irregular shape, and SR were key features in distinguishing ruptured cerebral aneurysms. Additionally, Kim et al. [29] constructed a Convolutional Neural Networks model for predicting IAs rupture risk. They trained the model with three-dimensional Digital Subtraction Angiography images. The AUC was 0.755.

How to accurately predict the risk of IAs rupture is clinically relevant, and the classification of IAs rupture status is still a challenge. Currently, most methods of IAs rupture risk assessment and prediction mainly adopted the geometrical parameters and hemodynamic parameters, without considering the deep features in hemodynamic clouds. Our experiments demonstrated the contribution of hemodynamic cloud features to IAs rupture risk prediction [30]. In this paper, a PointNet-based model was developed to extract different dimensional hemodynamic cloud features. Integrating IAs hemodynamic cloud features, geometrical parameters, and hemodynamic parameters, five types of machine learning algorithms were utilized to classify the rupture status of IAs. Furthermore, the impact of different dimensional hemodynamic cloud features on IAs rupture status classification was explored.

## 2. Materials and Methods

In this paper, an effective method for classifying IAs rupture status was proposed by integrating PointNet-based model and machine learning algorithms. Firstly, medical image segmentation and reconstruction algorithms were applied to process the 3D Digital Subtraction Angiography (DSA) data to construct three-dimensional IAs geometric models. IAs geometrical parameters were then determined with Geomagic, and hemodynamic clouds and hemodynamic parameters were determined with CFD. A PointNet-based model to extract hemodynamic cloud features was created; then, 1024-, 64-, 32-, and 16-dimensional hemodynamic cloud features were achieved. Finally, five types of machine learning algorithms were utilized to classify IAs rupture status by integrating geometrical parameters, hemodynamic parameters, and hemodynamic cloud features, and the effects of different dimensional hemodynamic cloud features on the classification of IAs rupture status were investigated. The method’s framework is illustrated in Figure 1.

### 2.1. IAs 3D Geometric Model

The geometric model of IAs is essential for measuring geometrical parameters and calculating hemodynamic parameters. Therefore, we first constructed patient-specific IAs geometric models based on clinical 3D Digital Subtraction Angiography (DSA) imaging data. The intracranial aneurysm imaging data were collected using the rotational DSA system (LCV; GE Medical Systems, Chicago, IL, USA) at Beijing Tiantan Hospital. The system performs a C-arm rotation before and after the injection of a contrast agent, acquiring a set of 44 3D images each time. The exposure rate was set at 8.8 frames per second, with each frame sized at 512 × 512, an inter-slice distance of 2.0 mm, a voxel size of 3.0 mm, and a DFOV of 2.10 cm. The GE workstation software (Advantage Unix; GE Medical Systems) compiled the acquired data into a DICOM (Digital Imaging and Communication in Medicine) format three-dimensional dataset, which was then imported into Mimics 13.0 (Belgium Materialize Company, Leuven, Belgium). Through level-set-based three-dimensional image segmentation and reconstruction, the vascular morphology of the intracranial aneurysm was modeled. Finally, it was converted to a 3D model of IAs in STL format, which was used as the original aneurysm vessel model for subsequent computational analysis. This study included 109 unruptured IAs and 40 ruptured IAs, yielding a total of 149 IAs 3D geometric models. The IAs 3D geometric models are shown in Figure 2. The raw imaging data for this study were provided by Beijing Tiantan Hospital, affiliated with Capital Medical University. All patients provided informed consent, and this protocol was approved by the hospital’s ethics committee.

### 2.2. IAs Geometrical Parameters

This study adopted 10 important geometrical parameters as classification features. The aneurysm sac height refers to the measurement from the highest point inside the sac to the base. It is primarily used to determine the size of the aneurysm. Other important measurements include the neck width, parent vessel diameter, AR, and SR. Based on the previously obtained geometric models of intracranial aneurysms, we used Geomagic (Raindrop Geomagic, Durham, NC, USA) to measure and calculate the corresponding parameters. The sac’s surface area, volume, and surface area to volume ratio (S/V) were obtained by CFD-Post calculations after subsequent hemodynamic analysis. Furthermore, there are two additional qualitative qualities. Some studies have suggested that the formation of daughter sacs can temporarily protect the aneurysm from rupture. Therefore, IAs were classified as the existence or absence of daughter sacs and were categorized as sidewall or bifurcation aneurysms based on the location of the growth.

### 2.3. IAs Hemodynamic Parameters

Using CFD methods, hemodynamic characteristics within the aneurysm and its parent vessel were calculated. The blood flow was simulated as a laminar flow of a viscous, non-compressible fluid that follows Newton’s laws of motion. The fluid has a density of 1055 kg/m^3^ and a viscosity coefficient of 0.004 kg/m·s. The governing equations consisted of the mass conservation equation and the momentum conservation equation. The boundary conditions were set as inlet velocity and outlet static pressure. It was assumed that the walls of the blood vessels were rigid and had no-slip properties. The inlet velocity was defined as a function of space and time, specified using a User Defined Function (UDF) in Fluent. Due to the non-slip condition at the vessel walls, the wall velocity was zero, creating a velocity gradient along the radial direction of the vessel. The blood flow velocity fluctuated in accordance with the cardiac cycle and could be quantified by the utilization of the Doppler ultrasonography technology. At the outlets, there was an absence of any pressure gradient. The simulations were conducted for two complete cardiac cycles, and the systolic portion of the second cycle was selected as the exportation. The duration of each cardiac cycle was 0.8 s, while the time interval for each cycle was 0.01 s. Hemodynamic analysis was conducted utilizing ANSYS Workbench Fluent (version 18.0, ANSYS Inc., Canonsburg, PA, USA).

After solving the control equations, the analysis of hemodynamic features was conducted using data from the systolic portion of the second cardiac cycle. CFD-Post allows for the visualization of the flow field conditions within the aneurysm and its parent vessel, including blood flow velocity, pressure on the vessel wall surface, WSS, OSI, and their distribution. It also generates corresponding hemodynamic clouds images, as illustrated in Figure 3. Following this, the hemodynamic clouds were extracted using CFD-Post from the surface of the patient’s intracranial aneurysm. Our extracted hemodynamic clouds contained 3000 points with multiple attributes on each point, such as the point’s 3D coordinates (x, y, z), WSS, OSI, pressure, and velocity. Finally, a total of 149 hemodynamic clouds were extracted and saved as a CSV file for the next step of extracting hemodynamic cloud features using the PointNet-based model.

Next, hemodynamic parameters were calculated. CFD-Post was used to calculate the High Oscillatory Shear Index area (HOA), Low Shear Area (LSA), WSS, OSI, pressure, velocity, and their respective maximum, average, and minimum values. In addition to these parameters, four qualitative indicators were added: whether the blood flow pattern was changed, whether the flow field was complex, whether the inflow jet was concentrated, and whether the blood flow impingement zone was concentrated. The blood flow pattern and flow field complexity were determined by observing flow streamline diagrams. Changes in the blood flow pattern referred to variations in the flow field within a cardiac cycle. If the flow field remained roughly unchanged over time, it was considered stable, denoted as 0. Conversely, changes in the flow pattern were marked as 1. A simple flow field was defined as having only one vortex, represented by 0, while a complex flow field, characterized by the presence of multiple vortices, was represented by 1. The region of flow impingement was determined by observing where the blood flow streamlines contact the inner surface of the aneurysm sac, and by assessing the distribution area of high WSS values (set as greater than 80% of the maximum WSS value on the sac surface). A value of 1 indicated a smaller, more concentrated impingement zone, while 0 indicated a broader, less concentrated impingement zone. The inflow jet could be defined according to the direction of the streamlines and the flow velocity. In this study, an inflow jet was defined as having significantly higher velocity compared to internal aneurysm flow streamlines, with a rapid decrease in speed and change in direction upon contact with the aneurysm’s inner surface. A more narrow and concentrated inflow jet was coded as 1, while a thicker, more dispersed inflow was coded as 0.

### 2.4. PointNet-Based Model to Extract Hemodynamic Cloud Features

A PointNet-based model to extract hemodynamic cloud features was created; the model structure is shown in Figure 4. The model could directly extract hemodynamic cloud features with different dimensions. From the extracted hemodynamic clouds, 1024 points were randomly selected as input for the model. The process began with inputting these points into a T-Net to obtain transformation parameters, which generated a transformation matrix. The original data were then multiplied by this matrix to align the point cloud, ensuring that the point cloud maintains rotational invariance. Subsequently, a shared-weight multilayer perceptron (MLP) was used to extract features from each point in the cloud, resulting in a corresponding feature matrix with dimensions increasing from 7 to C2. This step captured the local features of the point cloud. Another T-Net was then used to align these features, followed by another shared-weight MLP that gradually increased the feature dimensionality to C4. Finally, max pooling was performed to aggregate the features, outputting a 1 × C4 dimensional feature defined as the hemodynamic cloud features.

Hemodynamic cloud features of different dimensions were extracted by changing the sizes of the convolution kernels in the two shared MLP layers of the model. To be specific, this process could be achieved by changing the values of C1, C2, C3, C4 in Figure 4. When the model parameters (C1, C2, C3, C4) were set to 64, 64, 128, 1024, the hemodynamic cloud features extracted were 1024-dimensional. For settings of 8, 16, 32, 64, the cloud features were 64-dimensional; for 8, 8, 16, 32, the cloud features were 32-dimensional; and for 8, 8, 16, 16, the cloud features were 16-dimensional.

### 2.5. Machine Learning Algorithms

Five types of machine learning algorithms were utilized to classify the rupture status of IAs, as shown in Figure 5. And input features were categorized into two distinct groups, Group A and Group B. Group A contained only geometrical parameters and hemodynamic parameters; Group B consisted of Group A combined with different dimensions of hemodynamic cloud features. Group B was further divided into Groups B1, B2, B3, and B4, and the dimensions of hemodynamic cloud features employed by them were 1024, 64, 32, and 16, respectively. Each feature group was preprocessed with normalization before classification. Classification models were constructed using machine learning algorithms including XGBoost, CatBoost, Support Vector Machine (SVM), LightGBM, and Logistic Regression (LR). These models were applied to classify the rupture status of IAs applying Group A, Group B1, Group B2, Group B3, and Group B4 as feature inputs, respectively. Ten-fold cross-validation was employed to validate the models. The AUC, accuracy, sensitivity, and specificity of the models were calculated, and ROC curves were plotted.

## 3. Results

The constructed classification models were applied on Group A and Group B, respectively; for Group A, the accuracy for the XGBoost, CatBoost, SVM, LightGBM, and LR algorithms was 0.836, 0.816, 0.826, 0.832, and 0.885, respectively, with AUC values of 0.908, 0.922, 0.930, 0.884, and 0.921. For Group B, the accuracy for Group B1 was 0.846, 0.826, 0.816, 0.816, and 0.846, respectively, with AUC values of 0.904, 0.856, 0.773, 0.884, and 0.895. For Group B2, accuracy was 0.846, 0.836, 0.816, 0.826, and 0.885, respectively, with AUC values of 0.916, 0.908, 0.923, 0.902, and 0.935. For Group B3, accuracy was 0.857, 0.836, 0.836, 0.846, and 0.897, respectively, with AUC values of 0.926, 0.916, 0.928, 0.907, and 0.934. For Group B4, accuracy was 0.887, 0.857, 0.854, 0.857, and 0.908, respectively, with AUC values of 0.917, 0.934, 0.946, 0.920, and 0.944. Detailed results of the classification using various algorithms for Groups A and B are displayed in Table 1. The ROC curves are shown in Figure 6.

## 4. Discussion

This paper proposed an effective method for classifying the rupture status of IAs by integrating a PointNet-based model and machine learning algorithms. The PointNet-based model was created to extract hemodynamic cloud features, and 16-, 32-, 64-, and 1024-dimensional hemodynamic cloud features could be obtained by adjusting the C1, C2, C3, and C4 parameters in the model, respectively. The classification models of IA rupture status were developed utilizing five machine learning algorithms. The input parameters incorporated hemodynamic cloud features, geometrical parameters, and hemodynamic parameters. Furthermore, the effects of different dimensions of hemodynamic cloud features on the classification were explored.

This paper employed five common machine learning algorithms, each with its unique advantages, suited for different data characteristics and requirements. By collectively utilizing these algorithms, a comprehensive assessment and comparison of the classification performance for intracranial aneurysm rupture status were achieved. Among them, XGBoost and CatBoost are based on gradient boosting decision trees and are suitable for handling complex nonlinear relationships and high-dimensional data. SVM excels with small samples and high-dimensional data, particularly apt for precise classification boundaries. LightGBM offers advantages in processing large-scale data and speed of training. Logistic Regression is straightforward and interpretable, suitable for data with linear relationships.

In this study, Group B3 (Group A + 32-dimensional hemodynamic cloud features) and Group B4 (Group A + 16-dimensional hemodynamic cloud features) performed better than Group A across all five algorithms, with accuracies improved relative to Group A. Group B1 (Group A + 1024-dimensional hemodynamic cloud features) and Group B2 (Group A + 64-dimensional hemodynamic cloud features) only showed higher accuracies than Group A in the XGBoost and CatBoost algorithms, while SVM, LightGBM, and LR had lower accuracies compared to Group A. Additionally, the results indicated that as the dimensionality of the hemodynamic cloud features included decreased, the accuracies of all algorithms generally showed an upward trend, achieving the highest classification accuracies when the hemodynamic cloud features were 16-dimensional. This was likely because when the dimensionality of the hemodynamic cloud features was high, although more features were preserved, there were more redundant features. Due to the excessive number of features, samples became sparse in high-dimensional spaces, making distance measurement unreliable. This hindered the classifier’s ability to effectively distinguish between sample categories in high-dimensional spaces, thus deteriorating classification performance. Meanwhile, having too many features could lead to model overfitting, resulting in poorer performance on the test set. As the dimensionality of the hemodynamic cloud features decreased, redundant features were reduced, and it was easier for the model to find features with strong generalization ability, hence the overall improvement in accuracy.

Group B1 and Group B2 achieved higher accuracy than Group A only in the XGBoost and CatBoost algorithms. This may be attributed to the fact that XGBoost and CatBoost are gradient-boosting decision tree methods, which excel in handling high-dimensional and complex features, particularly effective with data that exhibit nonlinear relationships. These algorithms possess automated feature selection capabilities, enabling them to effectively select and combine features during the training process. This allows them to perform well even when dealing with datasets that contain many irrelevant or redundant features. In contrast, SVM, LightGBM, and LR may not perform as well in these situations.

In the classification results of Group B4, the LR algorithm achieved the highest accuracy at 0.908, while the SVM algorithm obtained the highest AUC at 0.946. This is likely because LR is a linear model, which is suitable for linearly separable datasets. When the feature dimension was moderate and the key features were prominent, LR effectively utilized these features for classification, thus enhancing accuracy. The 16-dimensional hemodynamic cloud features were of a moderate dimension that effectively captured key features related to aneurysm rupture status while avoiding the noise and redundancy associated with high-dimensional features. LR fitted the data well, thereby achieving the highest classification accuracy. On the other hand, SVM, with its strong discriminatory power and kernel functions, was able to find the optimal classification hyperplane when handling high-dimensional features, resulting in the highest AUC value. This outcome demonstrated the respective strengths of different algorithms when processing the same feature set, thereby justifying the study’s approach of combining five machine learning algorithms to comprehensively assess classification performance.

Compared to three-dimensional point clouds, the hemodynamic clouds contained more meaningful information. Hemodynamic clouds possessed multiple attributes, including not only morphological features but also hemodynamic characteristics. This suggested that hemodynamic clouds may uniquely contribute to the classification of IAs rupture status, a finding that was corroborated by existing research. The hemodynamic cloud features extracted using the PointNet-based model encapsulated morphological and hemodynamic characteristics relevant to the rupture status of IAs. Experiments demonstrated that adding 16-dimensional or 32-dimensional hemodynamic cloud features to the base of geometrical parameters and hemodynamic parameters effectively improved the classification of IAs rupture status. This provided new insights for predicting and assessing the risk of IAs rupture, suggesting that hemodynamic cloud features could serve as effective parameters for assessing IAs rupture risk, and the influence weights of different dimensions of hemodynamic cloud features on IAs rupture risk assessment were different. This method not only serves as an effective tool for predicting the rupture status of intracranial aneurysms but also offers new insights for predicting and assessing the rupture risks of thoracic and abdominal aortic aneurysms.

In this paper, CFD numerical simulations were adopted to obtain hemodynamic clouds and hemodynamic parameters, which may have some deviations from the actual hemodynamic characteristics. However, several studies have shown that CFD numerical simulation results have good consistency with actual hemodynamic characteristics [31,32]. Therefore, the CFD numerical simulations in this study are feasible.

This study still faced certain limitations. Firstly, the sample size of intracranial aneurysms was relatively small. Future research should include larger datasets of intracranial aneurysms, incorporating data from multiple medical centers to enhance the model’s generalizability and robustness. Additionally, the significant disparity in sample sizes between ruptured and unruptured IAs could lead to classification bias, weakening predictive accuracy for the less represented category. Furthermore, this study relied solely on CFD to obtain hemodynamic parameters. Although CFD is highly repeatable and efficient, it did not account for the deformation of the arterial walls in reality. Future research could consider using Fluid-Structure Interaction (FSI) to explore the impact of vascular wall deformation.

## 5. Conclusions

In this paper, an effective method for classifying IAs rupture status was proposed by integrating the PointNet-based model and machine learning algorithms. The 16-, 32-, 64-, and 1024-dimensional hemodynamic cloud features were extracted with the PointNet-based model, respectively, integrating hemodynamic cloud features, geometrical parameters, and hemodynamic parameters, and five types of machine learning algorithms were utilized to construct classification models for IAs rupture status. And the impact of different dimensional hemodynamic cloud features on the classification performance of IAs rupture status was compared. Experiments demonstrated that hemodynamic cloud features had a certain contribution weight to the classification of IAs rupture status, and it was found that adding 16-dimensional hemodynamic cloud features could effectively improve classification performance, with XGBoost, CatBoost, SVM, LightGBM, and LR algorithms achieving classification accuracies of 0.887, 0.857, 0.854, 0.857, and 0.908, respectively. This method could be an effective tool for predicting IAs rupture status. Our models and algorithms would provide valuable insights for the clinical diagnosis and treatment of IAs.

## Figures and Tables

**Figure 1 bioengineering-11-00660-f001:**
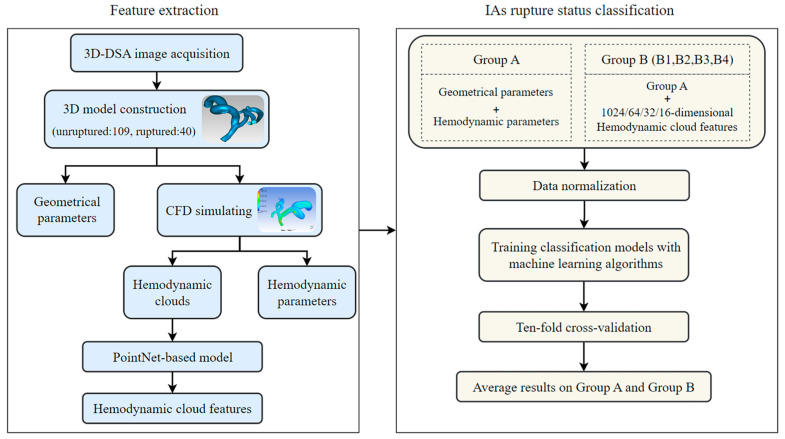
The computational framework for classification of IAs rupture status.

**Figure 2 bioengineering-11-00660-f002:**
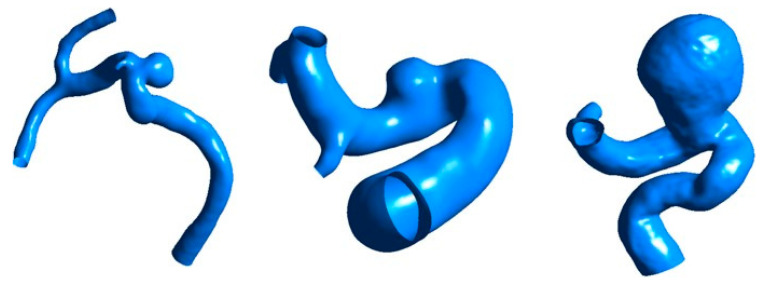
Three-dimensional geometric models of IAs.

**Figure 3 bioengineering-11-00660-f003:**
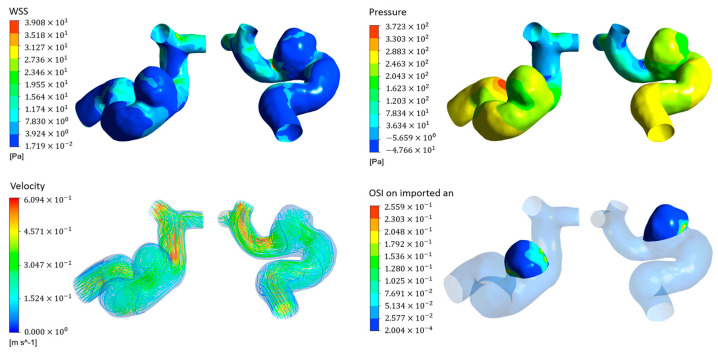
Hemodynamic clouds of WSS, pressure, velocity and OSI.

**Figure 4 bioengineering-11-00660-f004:**
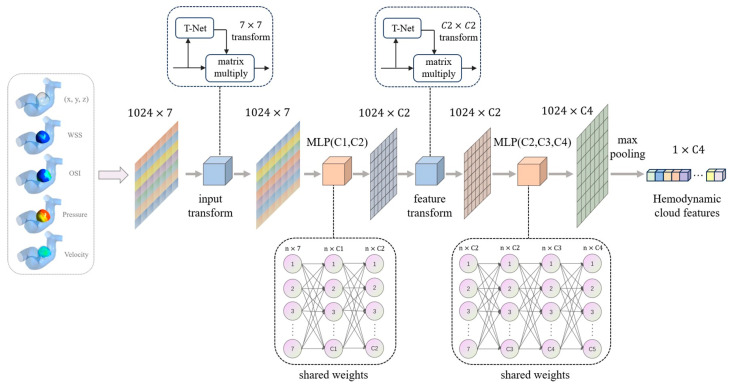
The architecture of PointNet-based model to extract hemodynamic cloud features.

**Figure 5 bioengineering-11-00660-f005:**
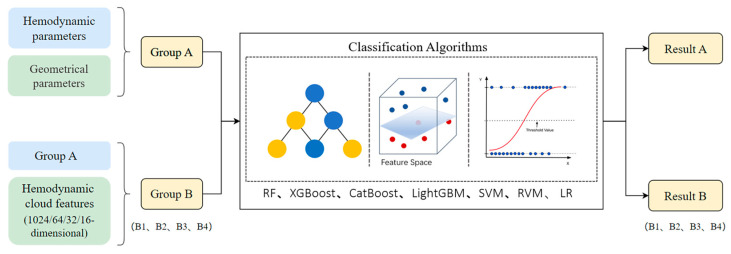
The computational framework for classification of IAs rupture status utilizing machine learning algorithms with different feature group inputs.

**Figure 6 bioengineering-11-00660-f006:**
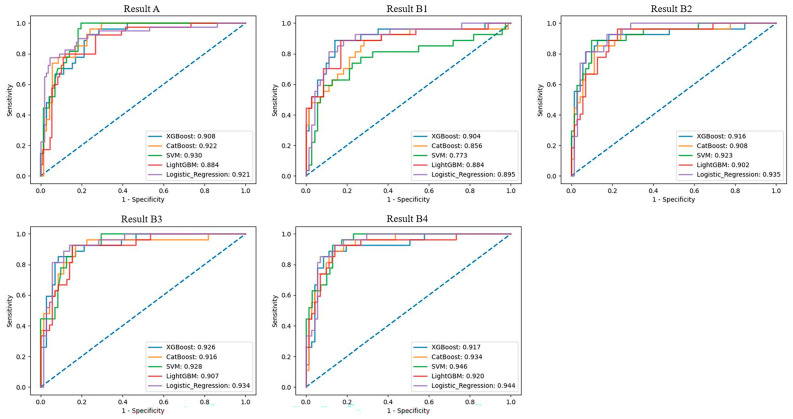
The ROC curves for Group A, B1, B2, B3, and B4 classification.

**Table 1 bioengineering-11-00660-t001:** Classification results of each group using different machine learning algorithms.

Algorithm	Feature	Accuracy	Sensitivity	Specificity	AUC
	Group A	0.836	0.592	0.929	0.908
	Group B1	0.846	0.592	0.943	0.904
XGBoost	Group B2	0.846	0.592	0.943	0.916
	Group B3	0.857	0.592	0.957	0.926
	Group B4	0.887	0.740	0.943	0.917
	Group A	0.816	0.481	0.943	0.922
	Group B1	0.826	0.444	0.971	0.856
CatBoost	Group B2	0.836	0.518	0.957	0.908
	Group B3	0.836	0.555	0.943	0.916
	Group B4	0.857	0.555	0.971	0.934
	Group A	0.826	0.518	0.943	0.930
	Group B1	0.816	0.592	0.901	0.773
SVM	Group B2	0.816	0.407	0.971	0.923
	Group B3	0.836	0.444	0.985	0.928
	Group B4	0.854	0.518	0.985	0.946
	Group A	0.832	0.600	0.917	0.884
	Group B1	0.816	0.518	0.929	0.884
LightGBM	Group B2	0.826	0.555	0.929	0.902
	Group B3	0.846	0.629	0.929	0.907
	Group B4	0.857	0.666	0.929	0.920
	Group A	0.885	0.775	0.926	0.921
	Group B1	0.846	0.703	0.901	0.895
LR	Group B2	0.885	0.777	0.927	0.935
	Group B3	0.897	0.814	0.929	0.934
	Group B4	0.908	0.851	0.929	0.944

## Data Availability

The data presented in this study are available on request from the corresponding author due to privacy concerns.

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
