# Peer review of "Integrating PointNet-Based Model and Machine Learning Algorithms for Classification of Rupture Status of IAs"

_bioengineering, 2024, doi:10.3390/bioengineering11070660_

Round 1

Reviewer 1 Report

Comments and Suggestions for Authors

The paper used machine learning tools to classify the rapture status of IAs.  Some details in methods are needed. 

What are the hemodynamic clouds?

Line 224. What are the data/features selected at the 1024 points? Why is 7 at the inputs (1024x7)

Comments on the Quality of English Language

There are some formatting issues. Remove extra spaces and hyphens.

Lines 119 and 350: change "kinds" to "types"

How was the t-net generated?

Table 1 needs horizontal lines to separate the groups for better view.

Reviewer 2 Report

Comments and Suggestions for Authors

This paper describes the proposal of an effective method for classifying the rupture status of intracranial aneurysms (IAs) by integrating PointNet-based Model and Machine Learning Algorithms.  As methods, three-dimensional IAs geometric models were constructed as by medical image segmentation and reconstruction algorithms and morphological variables of IAs were then acquired using Geomagic Geomagic software. In addition, the computation of hemodynamic cloud and hemodynamic parameters were also obtained by Computational Fluid Dynamics (CFD) and a PointNet-based model. As final process, 5 machine learning algorithms (XGBoost, CatBoost, SVM, LightGBM, and LR) were applied on morphological variables, hemodynamic parameters, and hemodynamic cloud features to classify and recognize the rupture status of IAs. The results shows that 16-, 32-, 64- and 1024-dimensional hemodynamic cloud features were extracted with the PointNet-based model. It is concluded that the models achieved the highest classification accuracies and AUCs when 16-dimensional hemodynamic cloud features of XGBoost, CatBoost, SVM, LightGBM, and LR algorithms were added to the morphological and hemodynamic features.

  It is very interesting topic for medical application to examine the methods to obtain the rupture status of intracranial aneurysms (IAs). But there are some questions and comments related your draft one as bellows. Please answer and update the draft paper.

1.      It is hard to find out your originality of this paper. All methods are known ones such as Machine learning or methods has been done by commercial software. There are many similar works to predict the risk of something about medical field. How about the originality from bioengineering point of view?

2.      In your conclusion, “the LR model achieved the highest classification accuracies and AUCs when 16-dimensional hemodynamic cloud features”. How do you propose a method including the ML part? Most of the readers will think that this is just a result, then you had better to explain why you can get these data from biomedical point of view. We are afraid that the results will depends on the data itself. In this case, your conclusion is not universal. Anyway please add the explanation related this matter in the discussion part, not result part.

3.      You did get hemodynamic parameter by using commercial CFD software, but how did get the validation of the flow itself. From our experiences about CFD, the flow pattern and related parameters such as pressure and shear stress field are more sensitive than velocities. Even if there are small differences between similar geometries,  the pressure and shear stress will be changed and proposed parameters will also changed. Then it is necessary to validate the CFD results themselves comparing with checking the geometries.

Anyway please explain about these validations or errors, and add them in the main text. This is very important process when CFD works will be used to the Machine Learning works.

Reviewer 3 Report

Comments and Suggestions for Authors

The study is concerned with the classification models for intracranial aneurysms (IAs). IAs rupture status were constructed by integrating PointNet-33 based model and machine learning algorithms. The application aspects of the study are sound.

The results of the experiments show that hemodynamic cloud features had a certain contribution weight to the classification of IAs rupture status. The models and algorithms used can be useful for clinical diagnostic purposes and treatment processes of IAs.

As suggestions, I can provide the following points to improve the paper:

The software used for conducting the analysis can be provided in the text. It can also be given among the references.

The novel aspects of the work can be provided by comparing the paper’s scheme and originality with other papers in the literature.

The contributions of the paper can be provided in further detail, please.

The applicability of the method in relation with other application areas could be provided, too.

It is stated that ensemble methods (as stated in the study: Classification models were constructed using machine learning algorithms including XGBoost, CatBoost, Support Vector Machine (SVM), LightGBM, and Logistic Regression (LR)) were used as well as machine learning methods. It could also be stated in the study why these particular methods were taken into consideration.

Future directions can be provided.

Yours faithfully,

Round 2

Reviewer 1 Report

Comments and Suggestions for Authors

Minor issue regarding the term "hemodynamic cloud".

Is the term "hemodynamic cloud" or "hemodynamic cloud"? Both were used in the paper.

Reviewer 2 Report

Comments and Suggestions for Authors

Thank you for your proper updating.
